# One-step fabrication of crystalline metal nanostructures by direct nanoimprinting below melting temperatures

Ze Liu[1]

Controlled fabrication of metallic nanostructures plays a central role in much of modern science and technology, because changing the dimensions of a nanocrystal enables tailoring of its mechanical, electronic, optical, catalytic and antibacterial properties. Here we show direct superplastic nanoimprinting (SPNI) of crystalline metals well below their melting temperatures, generating ordered nanowire arrays with aspect ratios up to $\sim 2,000$ and imprinting features as small as 8 nm. Surface-enhanced Raman scattering (SERS) spectra reveal strongly enhanced electromagnetic signals from the prepared nanorod arrays with sizes up to $\sim 100$ nm, which indicates that our technique can provide an ideal way to fabricate robust SERS substrates. SPNI, as a one-step, controlled and reproducible nanofabrication method, could facilitate the applications of metal nanostructures in bio-sensing, diagnostic imaging, catalysis, food industry and environmental conservation.

[1] Department of Engineering Mechanics, School of Civil Engineering, Wuhan University, Wuhan, Hubei 430072, China. Correspondence and requests for materials should be addressed to Z.L. (email: ze.liu@whu.edu.cn).

   1

Rapid and reproducible fabrication of metallic nanostructures is essential to much of modern science and technology[1-3], because changing the dimensions of a nanocrystal enables tailoring of its mechanical[4,5], electronic[6-8], optical[9-12], catalytic[2,13,14] and antibacterial properties[15]. It was found that the properties of metal nanostructures significantly depend on their sizes, shapes and aspect ratios[11,16,17], which has motivated an upsurge in research on developing process methods that allow better control of shape and size[15,18,19]. Chemical synthesis of metal nanoparticles has been well developed for preparing metal nanocrystals with good quality[20,21] since they were first documented by Faraday[22], but suffers from the limited selection of precursor compounds and the challenge of dispersing synthesized nanocrystals in liquids. The current preparation of metal nanopatterns mainly relies on advanced nanolithography techniques[3], such as nanosphere lithography[23,24] and electron beam lithography[25,26]. These methods allow the fabrication of homogeneously metallic nanopatterns but are costly, owing to time-consuming, multistep processes, and are also limited in preparing nanostructures with low-aspect ratios[27]. At present, rapid fabrication of metallic nanostructures in terms of controllability (for example, resolution, precision and uniformity), material diversity, cost and especially high-aspect ratio remains a significant challenge[28,29].

Among the variety of developed nanofabrication methods, nanoimprinting[30,31], pioneered by Chou et al.[30], promises reproducible fabrication of ordered and regular nanopatterns at great precision and at low costs but is only limited to polymers[3,31,32]. Recently, Schroers and co-workers[33-35] succeed in nanomoulding bulk metallic glasses (BMGs) above their glass transition temperatures ($T_g$) and below crystallization temperatures ($T_x$), which has triggered broad research in exploring BMGs' applications such as electrochemical catalysts[36] and micro/nano-electro-mechanical systems[37,38]. However, a similar process has been considered infeasible for nanoimprinting crystalline metals below their melting temperatures ($T_m$)[28,33,37] because of the limitations on formability originating from size effect in plasticity[4,39] and grain size effect[37,40]. There has been a few attempts using templates to nanoimprint metals close to forging[41], but the temperature is too low and the embossed smallest feature size is usually on the order of grain sizes. Few realized examples are through melting metals locally by laser heating[28,42].

Here we show direct nanoimprinting of a variety of crystalline metals (for example, Bi, Ag, Au, Cu and Pt) by superplastic forming well below $T_m$. This technique enables one step and rapid fabrication of metallic nanostructures with excellent controllability and predictability, in particular allowing the fabrication of high-aspect ratio nanostructures.

## Results

### Superplastic nanoimprinting of metals.
Figure 1a sketches the basic superplastic nanoimprinting (SPNI) steps for fabrication of crystalline metal nanostructures. A piece of metal is first placed onto a mould, where the commonly used moulds are $Al_2O_3$ templates prepared by anodic oxidation. When heated the metal/mould combinations to a target temperature, the metal is compressed against the mould and nanostructures form by materials creeping surrounding pores in the mould. The target temperature is usually set as $0.5T_m \leq T < T_m$, with unit of absolute scale of temperature. Typically, after SPNI, the mould is dissolved to release the replicated metal nanostructures. Figure 1b shows a typical as-thermoplastic-formed sample under an optical microscopy (9XB-PC, Shanghai optical instrument factory). The sample was fabricated by superplastically forming a piece of Au into a nanoporous $Al_2O_3$ template at 500 °C, well below the melting temperature of bulk Au ($T_m \sim 1,063$ °C). The $Al_2O_3$ template was purchased from Hefei Pu-Yuan Nano Technilogy Ltd. with an average pore diameter of 55 nm. The uniform rust-red colour of the sample in Fig. 1b suggests that Au nanowire arrays have been replicated, which is verified by characterizing the sample under a scanning electron microscopy (SEM, Zeiss Ultra Plus, Fig. 1c), after dissolving the $Al_2O_3$ template in KOH solution (concentration of 3 mol l$^{-1}$, temperature of 80 °C).

### Metal nanowires with controlled aspect ratio.
SPNI relies on direct mechanical deformation of mouldable materials and can therefore achieve great replicating precision in lateral dimensions. Now we investigate the controllability of SPNI along vertical direction. In general, the length of imprinted metal nanowire by SPNI is a function of forming pressure, processing temperature and time, which provides ways to control aspect ratio of replicated metal nanostructures. Taking nanoimprinting of Au as an example, the effect of holding time on aspect ratios of imprinted Au nanowires is first investigated. $Al_2O_3$ templates with uniform cylindrical nanopore sizes of $292 \pm 34$ nm are used in this experiment. First, five Au short rods ($61.8 \pm 0.5$ mg) were cut from an Au wire with a uniform diameter of 2 mm. Subsequently, the five Au short rods were superplastically formed into the $Al_2O_3$ templates at 500 °C, under an applied force of 10 kN and held for 1, 4, 8, 20 and 60 min, respectively. The lengths of Au nanowires at the centre of each sample are measured under SEM, which is used to calculate the aspect ratios of prepared Au nanowires (black dots in Fig. 2a). Theoretically, the length of replicated nanowire ($L$) under applied stress ($\sigma_s = (3S_{ij}S_{ij}/2)^{1/2}$, where $S_{ij}$ is the stress deviator) and temperature ($T$) can be generally quantified as

$$L = f(\sigma_s, T, t, d) \tag{1}$$

where $d$ and $t$ are diameter of nanopore and forming time, respectively. Applying classic Norton–Bailey's creep power law[43] gives

$$L = L_0 + A\exp\left(-\frac{Q}{RT}\right)\sigma_s^n t^m \tag{2}$$

where the constant $L_0$ approximates the lengths of metal flowing into nanopores during the loading force increasing from zero to the maximum value, and it is in general a function of temperature, nanopore size and loading rate. $R$ and $Q$ are gas constant and active energy for creep, respectively. $A$ is a constant related to nanopore size and material properties such as elastic modulus, $n$ is the stress exponent of creep rate. For nanoimprinting at constant temperature and constant force, for example, $T = 500$ °C and $F_m = 10$ kN in the above experiment, the calculated mean forming pressures ($\bar{\sigma} = F_m/S$, $S$ is the contact area between metal disc and the flat-end clamps of universal testing machine) for the five samples are $\bar{\sigma} = 471 \pm 24$ MPa (Supplementary Table 1), which can be approximated as a constant. Thus, the length and herein the aspect ratio of replicated nanowire will simply obey

$$L/d = B_1 + B_2 t^m \tag{3}$$

where $B_i$ ($i = 1, 2$) are constants related to nanopore size, material properties, processing temperature and forming pressure. Equation (3) indicates that the aspect ratio of prepared metal nanowire continues to increase as forming time increases, which can be explored to fabricate high-aspect ratio metallic nanostructures. This is very different from nanoimprinting of BMGs, where the processing time above $T_g$ is limited by the crystallization time due to the metastable nature of BMGs[33].

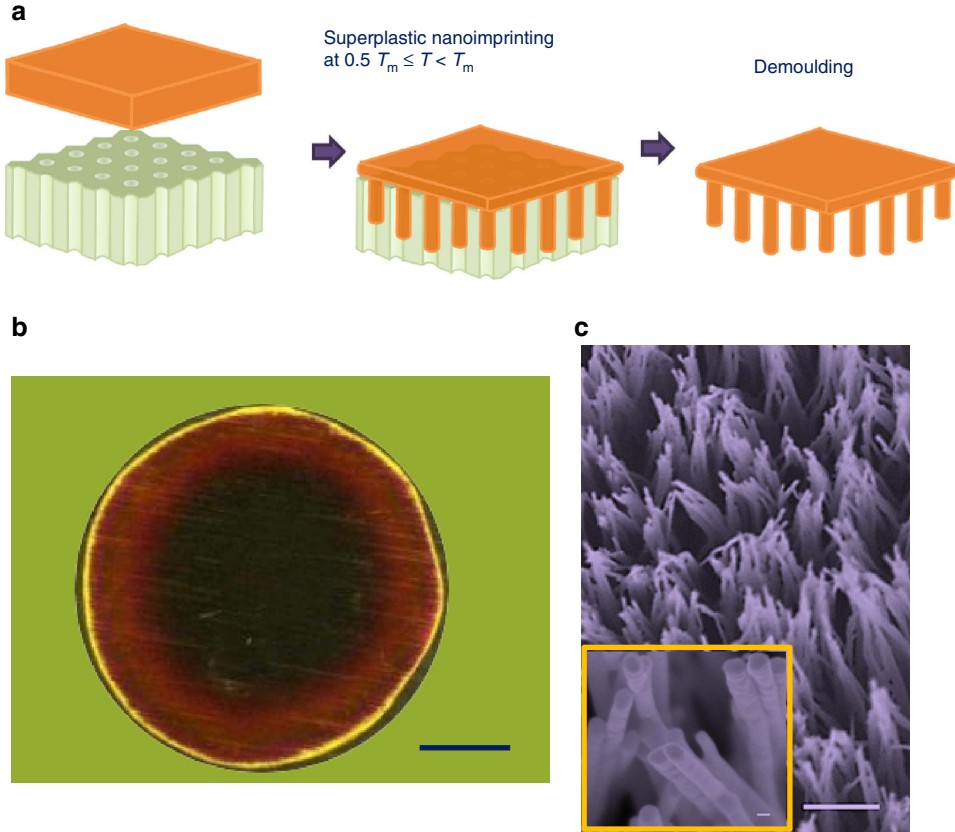

**Figure 1 | SPNI of crystalline metals below melting temperatures.** (**a**) Schematic of the SPNI process. (**b**) Optical micrograph of an as-thermoplastic-formed $Au/Al_2O_3$ template combination, which was prepared by SPNI under an applied force of 5 kN and holding for $\sim$60 min (Scale bar, 1 mm). The uniform rust-red colour of the sample suggests that Au nanowire arrays have been replicated, which is verified by characterizing the sample under SEM after dissolving the $Al_2O_3$ template in KOH solution ((**c**) scale bar, 5 μm. Inset figure scale bar, 30 nm).

One of the doubts about direct nanoimprinting of crystalline metals is the expected high strain rates to activate superplasticity in metals[28]. However, fitting of our experimental data by using equation (3) results in $L/d = 4.65 + 1.33 \times t^{0.36}$ (red line in Fig. 2a), which gives very low apparent strain rates $\left(\dot{\varepsilon}_{app} = d(\ln L)/dt\right)$, on the order of $10^{-3}\,s^{-1}$ in our experiments (Fig. 2b). The decreasing of $\dot{\varepsilon}_{app}$ as the length of nanowire increasing (black dots in Fig. 2b) is as anticipated, since the longer of nanowires, the higher of flow resistance force from wall friction. Further fitting to the $\dot{\varepsilon}_{app} - L$ data yields a power of $-4.45$ (red line in Fig. 2b), drastically deviating from the scaling law of Hagen–Poiseuille pipe flow, where $\dot{\varepsilon}_{app} \propto L^{-2}$ (see Methods section). Such a large discrepancy clearly rules out the viscous flow-dominated mechanism in the SPNI experiments.

In addition to varying holding time to control the length of replicated metal nanowires, processing temperature can also be independently varied to study its effect on the aspect ratios of prepared metal nanowires (Supplementary Fig. 1). For this purpose, seven Au short rods ($17.0 \pm 0.2$ mg) were first prepressed at 520 °C to get flat discs with thickness of $\sim$0.65 mm. Subsequently, the seven Au flat discs were superplastically formed into $234 \pm 37$ nm $Al_2O_3$ templates at 413, 463, 520, 558, 610, 713 and 773 °C, respectively, where all the samples are loaded to 1.5 kN and then held for 100 s. The measured lengths of Au nanorods at the centre of each sample are shown in Supplementary Fig. 1a. It is clear that the length of Au nanorod increases with the rising of temperature, which can be understood from the Arrhenius-type temperature relation in equation (2), the rising of temperature will increase the ability of the atoms to move. Further fitting of the experimental data with equation (2) gives $L/d = 4.50 + 9150.80 \exp(-5.48T_m/T)$ (red line in Supplementary Fig. 1a), based on which the creep activation energy in our experiment is calculated as $Q = 5.48 \times RT_m = 5.48 \times 8.31 \times (1063 + 273)\,J\,mol^{-1} = 14.5\,kcal\,mol^{-1}$, which is comparable to the measured gold surface diffusion activation energy $\sim$21 kcal per mole by using a scanning tunnelling microscope to monitor surface profiles decay at 125, 150 and 170 °C, respectively[44]. Such an agreement suggests that the creep deformation during SPNI is very possibly from diffusion-based mechanism.

**Replicate of metal nanoarchitecture and TEM characterization.** At present, fabrication of metal nanostructures smaller than 10 nm is very challenging, even with advanced nanolithography techniques. To demonstrate the powerful of this SPNI technique to replicate extremely small features, the smallest obtainable nanomould is adopted—a hierarchical $Al_2O_3$ template with multiple branched nanopores in its surface layer, where the branched nanopore sizes gradually increase from $\sim$8 nm in the outmost surface to 200 nm in the base layer (Supplementary Fig. 2). Figure 3a shows SEM images of replicated Au hierarchical nanostructures by using the hierarchical $Al_2O_3$ template (see Methods section). The aggregation of Au nanostructures into bundles makes it difficult to examine the replicated hierarchical structures. The prepared Au nanostructures were herein transferred onto a transmission electron microscopy (TEM) mesh grid (see Methods section). SEM imaging of Au hierarchical

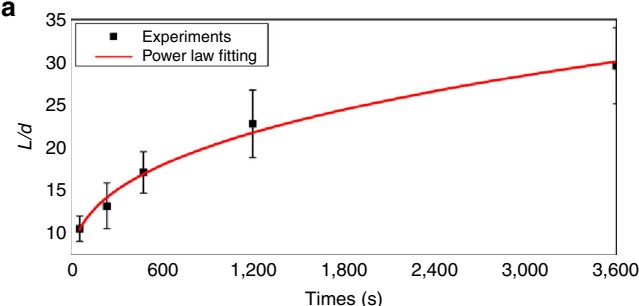

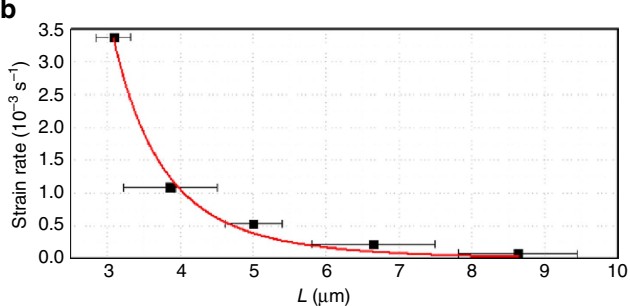

**Figure 2 | Fabrication of Au nanowire arrays with controlled aspect ratios by SPNI.** (a) The length of Au nanowires versus holding time, where five Au short rods with diameters of 2 mm (61.8 ± 0.5 mg) were superplastically formed into 200 nm $Al_2O_3$ templates at 500 °C, under an applied force of 10 kN and holding for 1, 4, 8, 20 and 60 min (black dots), respectively. The best fitting result by using equation (3) gives $L/d = 4.65 + 1.33 \times t^{0.36}$ (red line). (b) The calculated apparent strain rate ($\dot{\varepsilon}_{app} = d(\ln L)/dt$) in the experiments is on the order of $10^{-3}$ s$^{-1}$, and it is found scaling with $L$ as $\dot{\varepsilon}_{app} \propto L^{-4.45}$ by fitting with a power law (red line). The error bars are defined as s.d. and obtained by measuring at least 10 nanowires for each sample.

nanostructures on the TEM mesh grid clearly shows the primary stems abruptly multiplied by several small branches (Fig. 3b). The replicated smallest branch is ~8 nm (Supplementary Fig. 3), in consistent with the used hierarchical $Al_2O_3$ template, indicating the high replicating fidelity of the SPNI technique.

It is particularly noteworthy that the nanopore size at the entrance of the hierarchical $Al_2O_3$ template is only ~8 nm, three order of magnitude smaller than the grain size of the bulk Au (~$10^1$ μm, Supplementary Fig. 4), which suggests that the successful replication of metal nanostructures should originate from creep deformation of single crystals because most nanopores in the $Al_2O_3$ template are in contact with single crystals during SPNI. Considering that nanopores in the template will block the propagation of dislocations, called nanoscale geometrical confinement[4], the usual dislocation slip and twining mechanisms cannot work here. Direct evidence is from the characterization of Au hierarchical nanostructures under a high-resolution TEM (see Methods section)—there is no dislocations in the interior of replicated Au nanostructures and no observed slip steps on their surfaces (Fig. 4). Figure 4c shows a typical diffraction pattern for the selected Au nanostructure in Fig. 4b, which exhibits symmetrical and clear electron diffraction spots for single face-centered cubic crystal. The growing axis of the single crystal is determined along <111> crystallographic orientation. High-resolution TEM images (Fig. 4d–g) and fast Fourier transformations (insets of Fig. 4d–g) at selected regions denoted by A, B, C and D in the nanostructure in Fig. 4b confirm its perfect single crystallinity again. Most surprisingly, the crystallinity is not disturbed at all even around the regions where the branches abruptly converged (Fig. 4e,f).

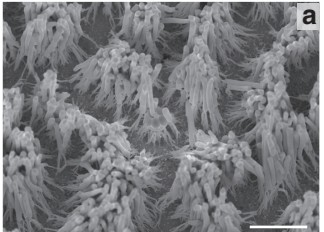
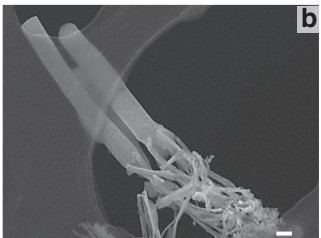

**Figure 3 | Au hierarchical nanostructures.** Au hierarchical nanostructures were replicated by superplastically forming a piece of Au into a hierarchical $Al_2O_3$ template with multiple branched nanopores in its surface layer at ~500 °C, under an applied force of 5 kN and holding for ~10 min. (a) SEM micrograph of fabricated Au hierarchical nanostructures. Scale bar, 1.5 μm. (b) Zoom-in imaging of Au hierarchical nanostructures on a TEM mesh grid (see Methods section) clearly shows the primary stems abruptly multiplied by several small branches. Scale bar, 100 nm.

**Discussion of mechanism.** Among the typical creep deformation mechanisms of viscous flow, dislocation motion, lattice and grain boundary diffusion, dislocation motion is attributed to the main creep mechanisms of crystalline metals at high temperatures, i.e. $T > 0.4T_m$ (ref. 45). In view of the fact that most of the metals possess grain sizes larger than 1 μm, direct nanoimprinting of crystalline metals below $T_m$ has been considered as impossible[28,33,37]. However, we noted that both viscous flow and lattice diffusion are independent of grains, which provides two possible mechanisms for direct nanoimprinting of metals. In fact, the viscous flow-based mechanism has been well explored to nanoimprint glasses[31,34] (that is, polymers and BMGs) above $T_g$ and below $T_x$, but it failed to explain our observations as discussed before (Fig. 2). We herein speculate that the creep mechanism during SPNI would originate from lattice diffusion. For lattice diffusion controlled creep mechanism, the strain rate can be estimated as ref. 45

$$\dot{\varepsilon} = A_1 \frac{D_s \mu b}{kT} \left( \frac{\sigma_s}{\mu} \right)^3 \qquad (4)$$

where $D_s$, $\sigma_s$, $\mu$, $b$ and $k$ are lattice diffusion coefficient, applied stress, shear modulus, magnitude of Burgers' vector and Boltzmann constant, respectively, $A_1$ is a dimensionless constant of order unity[45]. By substituting $D_s \sim 10$ nm$^2$ s$^{-1}$ (ref. 46) and elastic modulus $E = 70$ GPa (ref. 47) for Au at 500 °C, $\sigma_s \sim \bar{\sigma} = 471$ MPa in our experiment, typical values of Poisson's ratio $\nu = 0.3$ and $b \sim 0.1$ nm, $k = 1.38 \times 10^{-23}$ J K$^{-1}$ into equation (4), we obtain $\dot{\varepsilon} = 13 \times 10^{-3}$ s$^{-1}$, which is comparable but larger than our experimental results, indicating again that lattice diffusion is a reasonable mechanism to nanoimprint crystalline metals below $T_m$. Considering that higher strain rate from theoretical estimation (equation (4)) can result if dynamic recrystallization takes place[45], recrystallization is very likely to coincide with superplastic forming in our experiments, which, in turn, would explain why the imprinted Au nanostructures possess perfect single crystallinity (Fig. 4). We therefore conclude that the deformation mechanism of SPNI of crystalline metals well below $T_m$ and under the pressure smaller than 500 MPa could originate from lattice diffusion.

**Fabricating variety of metal nanostructures by SPNI.** The SPNI technique includes but not limited to fabricating Au nanos-tructures. To show the generalization of this SPNI technique for preparing crystalline metal nanostructures, we also fabricated Bi, Ag, Cu and Pt nanowires by SPNI (Fig. 5). Figure 5a shows replicated Bi nanowire arrays with aspect ratio of AR ~300, which is achieved by SPNI a piece of Bi at 260 °C, closing to its

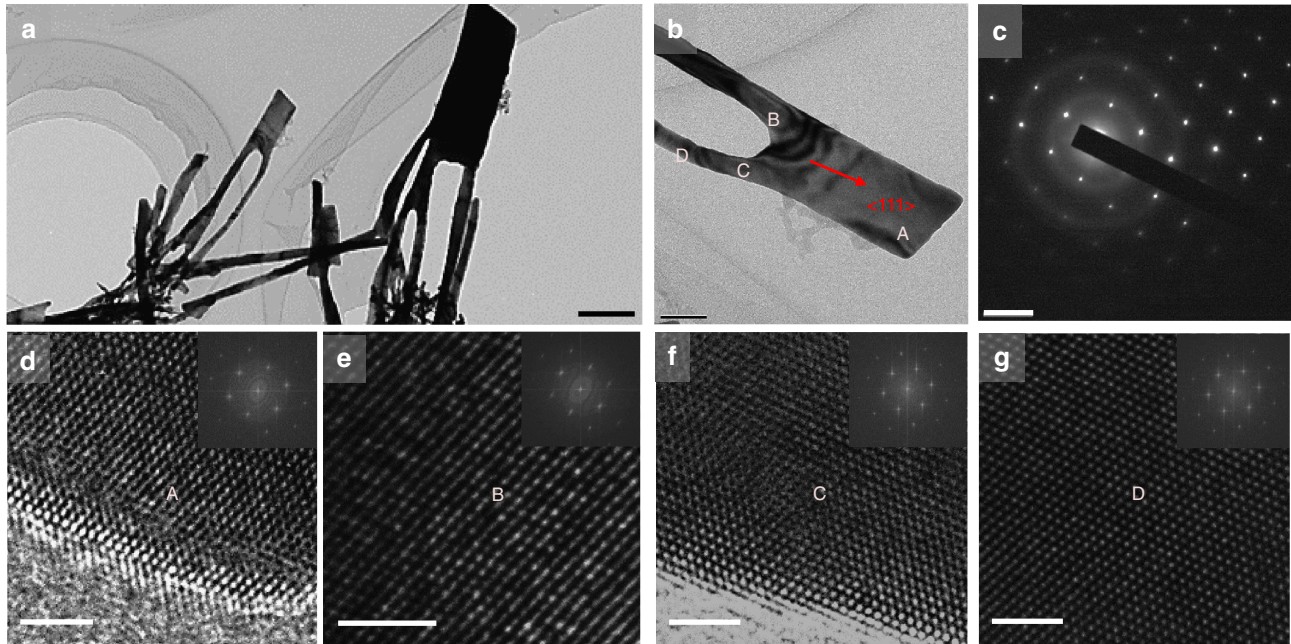

**Figure 4 | Characterization of Au hierarchical nanostructures by using TEM. (a,b)** Topography images of prepared Au nanostructures. Scale bars, 200 and 50 nm, respectively. (**c**) Diffraction pattern of the Au hierarchical nanostructure in **b** showing a face-centered cubic single crystal structure and the axis of the nanostructure is determined along <111> crystallographic orientation. Scale bar, 5 nm$^{-1}$. (**d-g**) High-resolution TEM images at the regions denoted by A, B, C and D in **b** and fast Fourier transformations (insets) confirming the perfect single crystal of the Au hierarchical nanostructure. Scale bars, 2 nm.

melting temperature ($T_m \sim 273\,°C$). It is observed that Bi can completely fill the template within 36 s under the applied force of 8 kN. Beside increasing processing temperature, extending forming time can also be applied to fabricate high-aspect ratio metal nanowires (as suggested by equation (3)). Figure 5b shows replicated Ag nanowires with extremely high-aspect ratios, up to $\sim 2,000$ (Fig. 5b), where the Ag nanowires almost fill up the $Al_2O_3$ template when holding at a forming pressure for $\sim 90\,min$ (the thickness of the $Al_2O_3$ template is $\sim 50\,\mu m$, see Supplementary Fig. 5). Figure 5c shows replicated Cu nanorod arrays at $550\,°C$, with aspect ratios lower than those of Ag nanowires by two–three orders of magnitude, which is understandable if estimating their strain rate difference. By substituting $\sigma_s \sim \bar{\sigma} \sim 500\,MPa$ in the experiments, $D_s = 5,090\,nm^2\,s^{-1}$, $\mu = 22\,GPa$ and $b = 0.286\,nm$ for Ag at $700\,°C$ (ref. 45), $D_s = 6\,nm^2\,s^{-1}$, $\mu = 38\,GPa$ and $b = 0.256\,nm$ for Cu at $550\,°C$ (ref. 45) into equation (4), we obtain $\dot{\varepsilon} = 12 \times 10^{-3}\,s^{-1}$ for nanoimprinting of Cu, which is similar to that of replicating Au nanorods but lower than the calculated strain rate for Ag by $\sim 3$ order of magnitude, where $\dot{\varepsilon} = 28\,s^{-1}$. The consistence between experiments and theoretical calculations indicate again that lattice diffusion is a reasonable mechanism for nanoimprinting of crystalline metals well below $T_m$.

In addition, it is worthwhile pointing out that bulk Pt, possessing melting temperature as high as $1,772\,°C$, can also be nanoimprinted at $\sim 820\,°C$ by using this method (Fig. 5d). What is more, the clear crystal facets and regular shapes observed at the top of Pt nanorods (Fig. 5d) indicate their excellent crystallinity. Similar crystallographic features are also observed in the imprinted Au nanorods with similar sizes (Supplementary Fig. 6). The above observations reveal one more advantage of SPNI, fabricating metal nanostructures with excellent crystalline quality.

**Surface-enhanced Raman scattering from Au nanowires.** Reproducible and robust metal nanostructures that strongly enhance the electromagnetic field are most desirable for surface-enhanced Raman scattering (SERS) but are difficult to achieve[29]. It is shown above that direct thermoplastic nanoimprinting of crystalline metals offers a rapid and controllable method to fabricate uniform metallic nanostructures, which could provide an ideal way to fabricate robust SERS substrates. To explore the effective of imprinted metal nanowire arrays as SERS substrates, we first prepared five Au flat discs, four of them were imprinted with Au nanowire arrays by SPNI and the remaining one is used as a reference sample (see Methods section). For simplicity, the reference sample is denoted as bulk Au while the four samples attached with Au nanowire arrays are denoted as s1, s2, s3 and s4, corresponding to nanowire sizes of 20, 55, 90 and 200 nm (Supplementary Fig. 7e–h), respectively. The optical micrographs of the as-thermoplastically formed $Au/Al_2O_3$ template combinations show clearly size-dependent colours (Supplementary Fig. 7a–d): the surface colour changes from pinkish (s1), rust-red (s2) to olive-green (s3) and finally tends to grey (s4). Such a absorption redshift is well understood from the size-dependent surface plasmon oscillation[17] and it indicates the effectiveness of the prepared samples as SERS substrates.

Supplementary Fig. 7i shows measured Raman signals at the centre of the prepared five samples by using $1.0 \times 10^{-5}\,M$ crystal violet (CV) as a sensitive SERS analyte to detect the electromagnetic enhancement (RENISHAW Raman microscope, INVIA, see Methods section). Although CV molecular absorbed on the bulk Au shows rather weak Raman signals, five Raman lines located at $442\,cm^{-1}$, $802\,cm^{-1}$, $1,174\,cm^{-1}$, $1,384\,cm^{-1}$ and $1,620\,cm^{-1}$ can still be recognized and these Raman shifts agree well with literature reports (Supplementary Table 2). On the contrary, almost all of the reported Raman lines for CV between 400 and $1,800\,cm^{-1}$ are drastically intensified by samples s1–s4 (Supplementary Fig. 7i and Supplementary Table 2), which demonstrates that the SPNI technique is very suitable to fabricate metallic SERS substrates. Supplementary Fig. 7i also shows that the Raman shifts decrease as the sizes of nanowires increase from

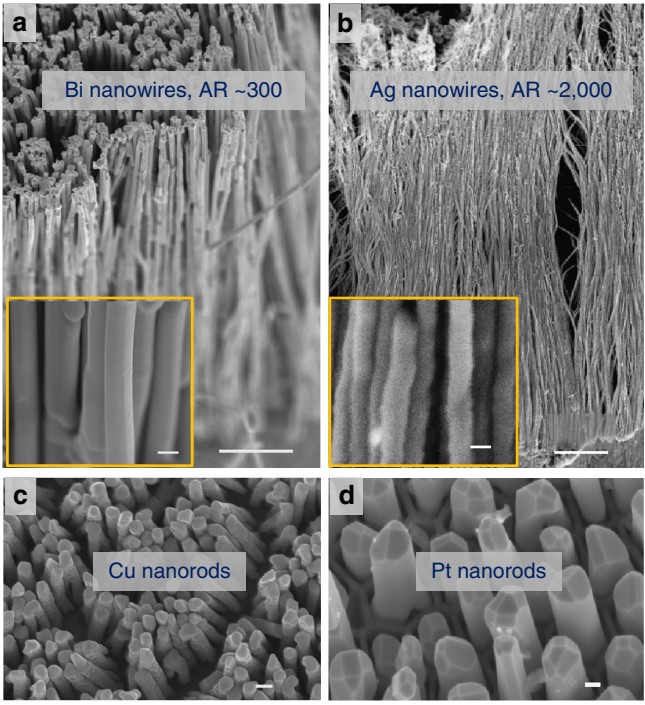

**Figure 5 | SEM images of nanowire arrays.** (**a**) SPNI a piece of Bi by using 200 nm $Al_2O_3$ template at 260 °C (closing to its melting temperature, $T_m$ ~273 °C) and under a force of 8 kN. Bulk Bi completely filled the $Al_2O_3$ template within 36 s, corresponding to an aspect ratio of ~300 since the thickness of $Al_2O_3$ template is ~60 μm. Scale bar, 5 μm. Inset figure scale bar, 200 nm. (**b**) An extremely high-aspect ratio of ~2,000 for Ag nanowires was also obtained by SPNI of a piece of Ag into 25 nm $Al_2O_3$ template at ~700 °C, under an applied force of 15 kN and holding for 90 min. Scale bar, 2 μm. Inset figure scale bar, 25 nm. (**c,d**) Cu and Pt nanowire arrays were fabricated by SPNI at ~550 and ~820 °C, respectively. Scale bars, 200 and 100 nm, respectively.

90 to 200 nm and to infinity (bulk Au). Such a size-dependent SERS signals is in good agreement with theoretical prediction[16,17]. However, when the sizes of nanowires continue to decrease below 90 nm, the Raman signals become slightly weaker rather than continuous increasing, which is attributed to the aggregation of small nanowires to form big bundles due to mechanical instability (Supplementary Fig. 7g–h).

## Discussion

Direct SPNI of crystalline metals well below $T_m$ has enabled to fabricate variety of metal nanowires with aspect ratios up to ~2,000. By adopting a hierarchical $Al_2O_3$ template with gradient nanopore size, Au hierarchical nanostructures are successfully replicated. The perfect monocrystalline structure of the prepared Au hierarchical nanostructures, together with the fact that the nanopore size at the entrance of the $Al_2O_3$ template is only ~8 nm, far smaller than the grain size of the bulk Au, we argue that direct SPNI of crystalline metals well below $T_m$ might originate from lattice diffusion-dominated mechanism, which is further supported by the agreement between theoretically calculated creep activation energy, strain rates and experimentally measured ones. Finally, the SPNI technique shows great advantage for fabricating metallic substrates for SERS application. SPNI is inherently reproducible due to direct contact deformation, and it requires only one step and simple equipment set-up, leading to low costs. We herein propose that this technique should facilitate the applications of metal nanostructures in catalysis, nanoelectronics, sensors and plasmonics.

## Methods

**Scaling of Hagen–Poiseuille flow.** Considering that a viscous fluid (viscosity of $\eta$) flows into a channel with diameter of $d$, under a constant driving pressure ($p_0$) at the entrance of the channel. Assuming the velocity of the fluid elements is so small that the fluid can be regarded as quasi-static flow, then the kinetic energy can be neglected and the energy dissipation rate from creeping-flow ($\dot{E}$) approximately equals to the power input by external forces ($\dot{W}$)

$$\dot{E} = \int_\Omega \sigma_{ij}\dot{\varepsilon}_{ij}d\Omega \qquad (5)$$

where $\sigma_{ij}$ and $\dot{\varepsilon}_{ij}$ are Cauchy stress tensor and rate of deformation tensor, respectively. The integral domain $\Omega$ is the fluid volume filled in the channel. By applying the constitutive law for incompressible Newtonian fluids[48],

$$\sigma_{ij} = -p\delta_{ij} + 2\eta\dot{\varepsilon}_{ij} \qquad (6)$$

where $p$ is hydrostatic pressure, we obtain

$$\dot{E} = \int_\Omega 2\eta\dot{\varepsilon}_{ij}^2 d\Omega \qquad (7)$$

For Hagen–Poiseuille flow, the only non-zero component is $\dot{\varepsilon}_{rz}$ (in cylindrical coordinates, $z$ is along the axis of the channel) and

$$\dot{\varepsilon}_{rz} = \frac{p_0 r}{4\eta L} \qquad (8)$$

where $L$ is the length of fluid filled in the channel. Substituting equation (8) into equation (7) gives

$$\begin{aligned}\dot{E} &= \int_\Omega 2\eta\left(\frac{p_0 r}{4\eta L}\right)^2 d\Omega \\ &= \frac{\pi p_0^2}{4\eta L^2}\int_0^l\int_0^{\frac{d}{2}} r^3 dr dz \\ &= \frac{\pi p_0^2 d^4}{256\eta L}\end{aligned} \qquad (9)$$

On the other hand, the power input by external forces (neglecting of body force) is

$$\dot{W} = F \cdot v = p_0 \frac{\pi d^2}{4} \cdot \frac{dL}{dt} \qquad (10)$$

By defining an apparent strain rate as

$$\dot{\varepsilon}_{app} = \frac{1}{L}\frac{dL}{dt} \qquad (11)$$

We finally obtain

$$\dot{\varepsilon}_{app} = \frac{p_0}{64\eta}\left(\frac{d}{L}\right)^2 \propto L^{-2} \qquad (12)$$

**Preparation of Au nanoarchitectures for TEM experiments.** A piece of bulk Au was brought into contact with an $Al_2O_3$ template with multiple branched nanopores in its outmost surface (Supplementary Fig. 2). After the $Au/Al_2O_3$ template combinations being heated to ~500 °C by using a furnace, the bulk Au was thermoplastically formed into the $Al_2O_3$ template under a constant loading rate of 0.05 mm per min until to a maximum force of 5 kN, then held at the maximum force for ~10 min. The imprinted nanostructures were subsequently demoulded from the $Al_2O_3$ template by dispersing the combinations in a KOH aqueous solution (concentration of 3 mol l$^{-1}$, temperature of 80 °C). The demoulded sample was then characterized under SEM (Fig. 3a) after rinsing with distilled water and acetone for five times. Finally, the replicated Au hierarchical nanostructures were released into 2 ml ethanol (concentration 100%) by sonication and subsequently transferred to a TEM mesh grid by using a micro pipette (Fig. 3b).

**SERS substrate fabrication.** CV with analytical purity purchased from Sinopharm Chemical Reagent Co., Ltd is used as the probe molecule to study the SERS activities of Au nanowire arrays. Before the SERS measurement, five Au short rods with the same volume were cut from an Au wire, four of them were superplastic formed into $Al_2O_3$ templates with nanopore sizes of 20 (s1), 55 (s2), 90 (s3) and 200 (s4) nm, respectively. The processing temperature is 500 °C and all of the four samples are held at 2.5 kN for 4 min. The remaining one was superplastically formed between two flat platens (without surface nanostructures) and set as the reference sample. After dissolving $Al_2O_3$ templates away and cleaning the samples with distilled water and acetone for several times, the five Au samples were finally immersed in the prepared ethanol with dissolved CV (concentration of $1.0 \times 10^{-5}$ mol l$^{-1}$) for ~30 min and then taken out to dry in the air.

**SERS measurement.** The SERS measurements were carried out at room temperature under a microscopic confocal Raman spectrometer (InVia Raman microscopes from Renishaw, England) by using a charge-coupled device detector

with a resolution of $1$–$2\,cm^{-1}$. The laser beam power is $17\,mW$ and only 15% of the power being used in the experiments to avoid fluorescence effect. Excitation wavelength of $514\,nm$, scan time of $30\,s$, field lens of 20 times and accumulation of 10 times were applied.

**Data availability.** The data that support the findings of this study are available from the corresponding author on reasonable request.

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

## Acknowledgements

This research was supported by the start funding from Wuhan University, the Fundamental Research Funds for the Central Universities, Hubei Provincial Natural Science Foundation of China (2016CFB159) and National Natural Science Foundation of China through Grant 11602175. The author is also grateful to the Analytical and Testing Center, Wuhan University for technical assistance.

## Author contributions

Z.L. designed and performed the experiments, analysed the data and wrote the paper.

## Additional information

**Competing interests:** The authors declare no competing financial interests.

