## [Peer Review File · Nature Communications]

Reviewers' comments:

Reviewer #1 (Remarks to the Author):

The paper "One-step fabrication of metal nanostructures by high-throughput imprinting" from Ze Liu presents the direct superplastic nanoimprinting (SPNI) of crystalline metals well below their melting temperatures. He was able to generate ordered nanowire arrays with aspect ratio up to ~ 2000 . The a very interesting observation of crystalline preserving imprint, accompanied by sound analysis of results using SERS and other techniques. The paper presents innovative work.

The author repeatedly talks about "we", but the paper only has one author. Are there some authors missing? Nor is there any acknowledgement to other collaborators.

In general, I would consider this work as good for publication in Nature, but I have some doubts whether both research and language is appropriate that it can be considered without major revision. While the text reads well, the language clearly needs some improvement, not only the English, but also many smaller gramatical and writing errors which seem to come from a huge haste in writing the paper.

About the research I have the impression that the real nature of the metal deformation and the lack of crystalline boundaries is still not completely understood. The author makes some attempts to explain his observations. There have been attempts to use nanoimprinting to pattern metals at room temperature or below the melting point, which is near to forging. Others melt metals locally by laser heating. Such references are missing. Futhermore the anodic alumina templates have been used in nanoimprinting, too. If the gold flows or creeps into nanopores, from where does the gold come from? In nanoimprint it comes from the compressed areas surrounding the pore, therefore it is not only a phenomena from inside the pores. Otherwise, it would be a kind of cutting. How about the volume conservation?

High throughput is a term often used if a mold is printed repeatedly into a moldable material. Here, high throughput means ability to upscaling to large area, fast (minutes to hours) pressing followed by dissolution of the master (KOH aqueous solution, heated at 80 °C for 2h). "Lost molds" are good for high reproducibility, but often not considered as high throughput.

While crystalline boundaries do not seem to form during molding, do they form after the demolding or after an annealing step?

Some more points:

Page 2: Not only shape and aspect ratio, but also size.

Page 2: Using the term "high throughput" would be acceptable if a repeated imprint is reported. Here it is a large area parallel imprint (also times of 30 min are quite large) in which the mold is lost after each imprint.

Page 3: Fluctuations of plasticity at the nanoscale? What is meant by this?

Page 5: What is "S"?

Page 5: Strain rate simular to that of $T > T_g$? (or better T_m , like in page 8?) Is there any T_g in metals?

Page 6: Temperature dependence - activity of atoms (not good term). Better: ability of the atoms to move.

Page 8: "it is quite possible that the deformation mechanism of superplastic nanoimprinting crystalline metals below T_m originates from lattice diffusion." The author should clarify the size of the pressure.

Page 8: is it now viscous flow or creep? If it is viscous flow, the sidewalls would play a significant role, which the author excluded due to the linear behavior.

Did the author take into account the ambient (oxygen) and a possible formation of an alloy with Al due to diffusion?

Use 5 kN instead of 5 KN (kilo)

What is "CV ethanol solution"?

Page 14: What is "Al₂O₃"? Do you mean Al₂O₃?

Reviewer #2 (Remarks to the Author):

In general, this work reports some useful discoveries that are of interest to the scientific community. One most attractive point is that the methodology seems applicable to many pure metals. The work can be considered for publication after some major revisions.

Major:

(1) A major weakness of this work is that the author does not rationalize the results quite well. There is a rich literature that is related to various high temperature creep mechanisms (Power-law creep, Herring-Nabarro creep, and Coble creep). The lattice diffusion based Power-law creep is most relevant to the reported observations. It is true that Norton-Bailey's equation is based on Power-law creep mechanisms. But the author should use the mechanism-based equation to figure out whether the estimated strain rate is reasonable (see <http://engineering.dartmouth.edu/defmech/>). The lattice diffusion coefficient, temperature, stress are all important variables. The Hagen-Poiseuille law might become relevant only when the superplastic temperature approaches the melting temperature of the metals (for example, in the case of Bi).

(2) In Fig. 2a, what does the author mean by "non-linear fitting"?

(3) Fig. 4 has a series of issues and is far from the publication quality. First, it appears that the diffraction spot distance of c and d is very different from those of a and b. Why is that? Second, b, d-f have two sets of scale bar labels. Third, Fig. 4c scale bar is not consistent with other panels.

(4) The nanowire quality is critical to the applications. Can the author add a zoomed-in TEM image in Fig. 1 to demonstrate the quality of Au nanowires? Extended Data Fig. 3 does show a HRTEM of the branched Au nanowire. But this image is in poor quality.

(5) In Fig. 5, it appears that Pt nanowires have much better crystal quality than Au. Can the author explain the reason(s)?

Minor:

(6) The reference style is inconsistent. Full journal names mix with abbreviations.

(7) Minor English grammar issues throughout the text.

[Editor's note: Please see attached file, which was provided by Reviewer #2 as a supplement]

Above $0.6T_M$ climb is generally *lattice-diffusion controlled*. The velocity v_c at which an edge dislocation climbs under a local normal stress σ_n acting parallel to its Burgers' vector is (Hirth and Lothe, 1968) [22]:

$$v_c \approx \frac{D_v \sigma_n \Omega}{bkT} \quad (2.17)$$

where D_v is the lattice diffusion coefficient and Ω the atomic or ionic volume. We obtain the basic climb-controlled creep equation by supposing that σ_n is proportional to the applied stress σ_s , and that the average velocity of the dislocation, \bar{v} , is proportional to the rate at which it climbs, v_c . Then, combining eqns. (2.2), (2.3) and (2.17) we obtain:

$$\dot{\gamma} = A_1 \frac{D_v \mu b}{kT} \left(\frac{\sigma_s}{\mu} \right)^3 \quad (2.18)$$

where we have approximated Ω by b^3 , and incorporated all the constants of proportionality into the dimensionless constant, A_1 , of order unity.

Reviewer #3 (Remarks to the Author):

It is an interesting paper which considers challenging issues of surface architecturation of metals. It must be mentionned that quite similar processes have been widely reported in litterature and the authors could probably mention in more détails these papers (see for instance papers published by Schroers and co workers). However, these process were rarely applied with the same success for polycrystalline materials. The results shown in this manuscript are consequently of interest for the community. Some scienfitic aspects related to the interpretation could be improved since in the present version of the manuscript, the involved mechanism of deformation is not clearly established. For instance the authors seem to oppose viscous flow and lattice diffusion as possible mechanism of deformation during nanoimprinting. The reviewer suggests to clarify what the authors want to mention. In order to identify the mechanism of deformation (which is probably lattice diffusion), estimation of the sensitivity to temperature via for instance the calculation of an activation energy (under appropriate assumptions) could be of interest.

Response to Editor/Reviewer

Many thanks for the Editor/Reviewer's valuable reports, which have further helped improve this paper. I have taken all the comments into consideration and have made appropriate changes to the manuscript in order to address them. My point-by-point responses are explained in detail below. In addition, changes to the revised manuscript are highlighted in blue.

Reviewer #1 (Remarks to the Author):

The paper "One-step fabrication of metal nanostructures by high-throughput imprinting" from Ze Liu presents the direct superplastic nanoimprinting (SPNI) of crystalline metals well below their melting temperatures. He was able to generate ordered nanowire arrays with aspect ratio up to ~2000. The a very interesting observation of crystalline preserving imprint, accompanied by sound analysis of results using SERS and other techniques. The paper presents innovative work. In general, I would consider this work as good for publication in Nature, but I have some doubts whether both research and language is appropriate that it can be considered without major revision.

Response: I very appreciate the reviewer's encouraging comments.

The author repeatedly talks about "we", but the paper only has one author. Are there some authors missing? Nor is there any acknowledgement to other collaborators.

Response: "we" has been changed to "I" and I have added "The author is also grateful to the Analytical and Testing Center, Wuhan University for technical assistance." in the acknowledgement.

While the text reads well, the language clearly needs some improvement, not only the English, but also many smaller gramatical and writing errors which seem to come from a huge haste in writing the paper.

Response: I have tried my best to improve the language and correct grammar and writing errors.

About the research I have the impression that the real nature of the metal deformation and the lack of crystalline boundaries is still not completely understood. The author makes some attemps to explain his observations. There have been attemps to use nanoimprinting to pattern metals at room temperature or below the melting point, which is near to forging. Others melt metals locally by laser heating. Such references are missing. Futhermore the anodic alumina templates have been used in nanoimprinting, too.

Response: The high temperature creeping mechanism is very complicated, but based on my observations, the new estimated strain rate from lattice diffusion and calculated creep activation energy in revised manuscript (See page 7 and 9 or my responses to point 1 by reviewer #2 and the last point by reviewer #3), I conclude that lattice diffusion is quite

possible the behind mechanism in my experiments. In addition, more references are added as the reviewer suggested (see also page 3): “There are a few attempts to use templates to nanoimprint metals near to forging⁴¹, but the temperature is relatively too low and the embossed smallest feature size is usually on the order of grain sizes. Few realized examples were through melting metals locally by laser heating^{28,42}.”

If the gold flows or creeps into nanopores, from where does the gold come from? In nanoimprint it comes from the compressed areas surrounding the pore, therefore it is not only phenomena from inside the pores. Otherwise, it would be a kind of cutting. How about the volume conservation?

Response: I described more about the superplastic forming process as following (See also page 3): “Figure 1a sketches the basic superplastic nanoimprinting steps for fabrication of crystalline metal nanostructures. A piece of metal is firstly placed onto a mold, where the commonly used molds are Al₂O₃ templates prepared by anodic oxidation. When heated the metal/mold combinations to a target temperature, the metal is compressed against the mold and nanostructures form by materials creeping surrounding pores in the mold.” The volume change during plastic deformation at high temperatures is insignificant so volume conservation is generally considered to be maintained (e.g. Ref. 44 in the revised manuscript).

High throughput is a term often used if a mold is printed repeatedly into a moldable material. Here, high throughput means ability to upscaling to large area, fast (minutes to hours) pressing followed by dissolution of the master (KOH aqueous solution, heated at 80 °C for 2h). "Lost molds" are good for high reproducibility, but often not considered as high throughput.

Response: “High throughput” has been changed to “high reproducible”.

While crystalline boundaries do not seem to form during molding, do they form after the demolding or after an annealing step?

Response: The reviewer’s finding is very interesting. Recrystallization is very possible to coincide with superplastic forming here since there is no additional annealing step in my experiments and the demolding temperature is also very low (80°C). In fact, creep accompanied by dynamic recrystallization usually takes place at high temperatures (Ref. 44 in the revised manuscript). See added discussions at page 10.

Some more points:

Page 2: Not only shape and aspect ratio, but also size.

Response: “It was found the properties of metal nanostructures significantly depend on their shapes and aspect ratios” has been changed to “It was found the properties of metal nanostructures significantly depend on their sizes, shapes and aspect ratios” (See also page 2).

Page 2: Using the term "high throughput" would be acceptable if a repeated imprint is

reported. Here it is a large area parallel imprint (also times of 30 min are quite large) in which the mold is lost after each imprint.

Response: “high throughput” has been changed to “high reproducible” as the reviewer suggested.

Page 3: Fluctuations of plasticity at the nanoscale? What is meant by this?

Response: Fluctuation of plasticity at the nanoscale means non-uniform plastic deformation, which results from size effect and is caused by localized dislocation bursts (Uchic M. D., et al., *Annu. Rev. Mater. Res.*, 39, 361-386, 2009). “Fluctuations of plasticity at the nanoscale” has been deleted to avoid confusions (See also page 3).

Page 5: What is "S"?

Response: I thanks to the reviewer for carefully reading the manuscript. *S* is the contact area between metal disc and the flat-end clamps of universal testing machine (See also page 5).

Page 5: Strain rate similar to that of $T > T_g$? (or better T_m , like in page 8?) Is there any T_g in metals?

Response: There is no T_g (glass transition temperature) in crystalline metals. I am sorry for the misleading by using abbreviation here, “BMGs” has been changed to “bulk metallic glasses”. Nanoimprinting, pioneered by Prof. Chou (Ref. 33), has been mainly applied in glasses such as polymers and metallic glasses (Refs. 3, 33-36) because they can drastically soften (reduction of viscosity) and hence easy to deform when heated above T_g but below T_x . Schroers and co-workers firstly succeeded in nanoimprinting of bulk metallic glasses above T_g and below T_x (See also pages 2-3).

Page 6: Temperature dependence - activity of atoms (not good term). Better: ability of the atoms to move.

Response: “the rising of temperature will increase the activity of atoms and/or defects” has been changed to “the rising of temperature will increase the ability of the atoms to move” (See also page 7).

Page 8: "it is quite possible that the deformation mechanism of superplastic nanoimprinting crystalline metals below T_m originates from lattice diffusion." The author should clarify the size of the pressure.

Response: I have modified as “it is quite possible that the deformation mechanism of superplastic nanoimprinting crystalline metals below T_m and under the pressure < 500 MPa in my experiments originates from lattice diffusion.” (See also page 10).

Page 8: is it now viscous flow or creep? If it is viscous flow, the sidewalls would play a significant role, which the author excluded due to the linear behavior.

Response: I am sorry for the misleading and I have modified the discussions a lot (See page 9 or my response to point 1 by reviewer #2). As the reviewer noted, the sidewalls would

play significant role for viscous flow, which has been excluded based on my experimental observations (Fig. 2).

Did the author take into account the ambient (oxygen) and a possible formation of an alloy with Al due to diffusion?

Response: The reviewer's concerns are very crucial. Oxidation is a key problem when processing metals at high temperature and in the ambient condition. This is why the author mainly selected noble metals (e.g. Au, Ag and Pt) to do nanoimprinting. For Cu nanowires, I indeed observed that the surface quality is not very good (Fig. 5c), which should result from the oxidation of Cu during SPNI.

For the question of "possible formation of an alloy with Al due to diffusion?" I think it is not very possible to form an alloy with Al because Al_2O_3 is very stable at high temperature (e.g. $< 600^\circ\text{C}$). Even if there is an Al alloy formed in the surface of molded metal nanowires, Al would be easily dissolved in KOH solution during following demolding process.

Use 5 kN instead of 5 KN (kilo)

Response: "KN" has changed to "kN".

What is "CV ethanol solution"?

Response: CV is the abbreviation of crystal violet (a chemical), I dissolved CV in ethanol. "CV ethanol solution" has changed to "ethanol with dissolved CV" (See also page 16).

Page 14: What is "Al₂O₃"? Do you mean Al_2O_3 ?

Response: I am sorry for the writing error, "Al₂O₃" has changed to " Al_2O_3 " (See also page 16).

Reviewer #2 (Remarks to the Author):

In general, this work reports some useful discoveries that are of interest to the scientific community. One most attractive point is that the methodology seems applicable to many pure metals. The work can be considered for publication after some major revisions.

Response: I appreciate the reviewer's encouraging comments. I believe the manuscript has improved a lot by taking the reviewer's valuable concerns into consideration.

Major:

(1) A major weakness of this work is that the author does not rationalize the results quite well. There is a rich literature that is related to various high temperature creep mechanisms (Power-law creep, Herring-Nabarro creep, and Coble creep). The lattice diffusion based Power-law creep is most relevant to the reported observations. It is true that Norton-Bailey's equation is based on Power-law creep mechanisms. But the author should use the mechanism-based equation to figure out whether the estimated strain rate is reasonable (see <http://engineering.dartmouth.edu/defmech/>). The lattice diffusion coefficient, temperature, stress are all important variables. The Hagen-Poiseuille

law might become relevant only when the superplastic temperature approaches the melting temperature of the metals (for example, in the case of Bi).

Response: The reviewer's suggestion is very helpful. I also believe that the lattice diffusion based creep mechanism is most relevant to my observations. According to the reviewer's kindly attached document, I estimated the strain rate from theory and compared with my experimental results as follows (See also page 9).

“For lattice diffusion controlled creep mechanism, the strain rate can be estimated as ⁴⁴

$$\dot{\epsilon} = A_1 \frac{D_s \mu b}{kT} \left(\frac{\sigma_s}{\mu} \right)^3 \quad (4)$$

where D_s , σ_s , μ , b and k are lattice diffusion coefficient, applied stress, shear modulus, magnitude of Burgers' vector and Boltzmann constant respectively, A_1 is a dimensionless constant of order unity ⁴⁴. By substituting $D_s \sim 10^{-13} \text{ cm}^2/\text{s}$ ⁴⁵ and elastic modulus $E = 70 \text{ GPa}$ ⁴⁶ for Au at 500°C , $\sigma_s \sim \bar{\sigma}_s = 471 \text{ MPa}$ in the experiments, typical values of Poisson's ratio $\nu = 0.3$ and $b \sim 0.1 \text{ nm}$, $k = 1.38 \times 10^{-23} \text{ J/K}$ into eq. 4, I obtain $\dot{\epsilon} = 13 \times 10^{-3} \text{ s}^{-1}$, which is larger than the experimental results, indicating **again** (I calculated creep activation energy in my experiments at page 7 and it is found **also** comparable to the gold surface diffusion activation energy. See my response to the last point by reviewer #3) that lattice diffusion is a reasonable mechanism to nanoimprint crystalline metals below T_m . Considering that the higher of strain rate from theoretical estimation (eq. 4) can occur if dynamic recrystallization takes place ⁴⁴, recrystallization is very likely to coincide with superplastic forming in my experiments, which agrees with the observed perfect single crystallinity of the imprinted Au nanostructures (Fig. 4).”

(2) In Fig. 2a, what does the author mean by “non-linear fitting”?

Response: “non-linear fitting” has been changed to “power law fitting”.

(3) Fig. 4 has a series of issues and is far from the publication quality. First, it appears that the diffraction spot distance of c and d is very different from those of a and b. Why is that?

Response: I have improved the quality of the figure (See page 25). The diffraction spot distance of c and d is actually the same, such a visual illusion is mainly from the low quality of high-resolution TEM images because the size of the nanowire is a little big for TEM to pass electrons through, but the spots directions may rotate a little bit due to slightly bending of the slender nanowire. In fact, the diffraction pattern from the whole Au hierarchical nanostructure clearly shows its perfect monocrystalline nature (Fig. 4b-c).

Second, b, d-f have two sets of scale bar labels.

Response: The figures b-g are original images, where the scale bars are clear but the scale bar labels are too small to see clearly, so I input a new set of big labels in the figures. I have deleted one set of bar labels in the revised manuscript (See page 25).

Third, Fig. 4c scale bar is not consistent with other panels.

Response: The reviewer is right, the scale bar in Fig. 4c is not consistent with other panels because it is the diffraction pattern in reciprocal space, where the unit of the scale bar is 1/nm.

(4) The nanowire quality is critical to the applications. Can the author add a zoomed-in TEM image in Fig. 1 to demonstrate the quality of Au nanowires? Extended Data Fig. 3 does show a HRTEM of the branched Au nanowire. But this image is in poor quality.

Response: I have replaced Fig. 4d-g with zoomed-in TEM images with good quality.

(5) In Fig. 5, it appears that Pt nanowires have much better crystal quality than Au. Can the author explain the reason(s)?

Response: The crystal quality of Au is also very good, which can be seen from the new TEM images with improved quality and the diffraction pattern in Fig. 4c. The different topographies between Fig. 1c (Au) and Fig. 5d (Pt) are mainly from the size difference - it is challenge for SEM to see edges between crystal planes when the size of nanorod is very small (e.g. ~ 30 nm in Fig. 1c). For large size of Au nanorods arrays (with diameter similar to Pt nanorods in Fig. 5d), I do see clear crystal facets and regular shapes at the top of Au nanorods (See image below or new added Extended Data Fig. 6 in the revised manuscript).

Minor:

(6) The reference style is inconsistent. Full journal names mix with abbreviations.

(7) Minor English grammar issues throughout the text.

Response to (6) and (7): Thanks to the reviewer for so carefully reading the manuscript, I have corrected the inconsistent reference style and English grammar issues.

[Editor's note: Please see attached file, which was provided by Reviewer #2 as a supplement]

Response: Thanks to the editor for kindly remind me of this.

Reviewer #3 (Remarks to the Author):

It is an interesting paper which considers challenging issues of surface architecturation of metals. It must be mentioned that quite similar processes have been widely reported in literature and the authors could probably mention in more details these papers (see for instance papers published by Schroers and co-workers). However, these process were rarely applied with the same success for polycrystalline materials.

Response: I appreciate the reviewer's encouraging comments. More details about the applying of nanoimprinting in bulk metallic glasses are added (see pages 2-3). "...Recently, Schroers and co-workers firstly succeeded in nanomoulding of bulk metallic glasses (BMGs) above their glass transition temperatures (T_g) and below crystallization temperatures (T_x)³³⁻³⁵, which has triggered a broad of researches in exploring BMGs' applications such as electrochemical catalysts³⁶ and MEMS/NEMS^{37,38}. However, similar process has been considered infeasible for nanoimprinting crystalline metals below their melting temperatures (T_m)^{28,33,37} because of ... There were a few attempts using templates to nanoimprint metals...".

The results shown in this manuscript are consequently of interest for the community. Some scientific aspects related to the interpretation could be improved since in the present version of the manuscript, the involved mechanism of deformation is not clearly established. For instance the authors seem to oppose viscous flow and lattice diffusion as possible mechanism of deformation during nanoimprinting. The reviewer suggests to clarify what the authors want to mention.

Response: I have added more discussions on the mechanism as follows (See also pages 8-9): "...In view of the fact that most of metals possess grain sizes larger than 1 μm , direct nanoimprinting of crystalline metals below T_m has been considered as impossible^{28,33,37}. However, I noted that both viscous flow and lattice diffusion are independent of grains, which provides two possible mechanisms for direct nanoimprinting of metals. The viscous flow based mechanism has been well explored to nanoimprint glasses^{31,34} (i.e. polymers and BMGs) above T_g and below T_x , but it is failed to explain my observations as discussed before (Fig. 2). I herein speculate that the creep mechanism in my experiments originates from lattice diffusion..."

Based on the fact that both estimated strain rate and calculated activation energy from diffusion creep are in agreement with my experiments (See pages 7 and 9 or my responses to point 1 by reviewer #2 for strain rate estimation and my response below for activation energy calculation), I get to the conclusion: it is quite possible that the deformation mechanism of superplastic nanoimprinting of crystalline metals below T_m and under the pressure < 500 MPa in my experiments originates from lattice diffusion.

In order to identify the mechanism of deformation (which is probably lattice diffusion), estimation of the sensitivity to temperature via for instance the calculation of an activation energy (under appropriate assumptions) could be of interest.

Response: I thank the reviewer for this insightful comment. By calculating the activation energy, I found it is very comparable to the reported gold surface diffusion activation

energy. I added more discussions in the manuscript as follows (See also page 7) “...Further fitting the experimental datum with eq. (2) gives $L/d = 4.50 + 9150.80 \exp(-5.48T_m/T)$ (Red line Extended Data Fig. 1a), based on which the creep activation energy in my experiments is calculated as $Q = 5.48 \times RT_m = 5.48 \times 8.31 \times (1063 + 273)$ J/mol = 14.5 kcal/mol, which is comparable to the measured gold surface diffusion activation energy ~21 kcal/mol by using a scanning tunneling microscope to monitor surface profiles decay at 125, 150 and 170 °C, respectively ⁴³. Such an agreement suggests that the creep deformation in my experiments is very possibly from diffusion based mechanism.”

REVIEWERS' COMMENTS:

Reviewer #1 (Remarks to the Author):

I appreciate very much the amendments made to the manuscript based on my review and also on those of the other reviewers. I consider the corrections and additions sufficient and suggest a publication of the paper as it is.

Reviewer #2 (Remarks to the Author):

Two additional comments are offered:

- (1) The use of "I" is rather awkward. This can be easily rectified with a passive voice.
- (2) The author only estimated the deformation/strain rate of Au nanowires based on lattice diffusion equation. What about Ag, Cu, and Pt? It is important to have a self-consistent story.

Response to Reviewer

Many thanks for the Reviewer's remarks, which I have taken into consideration and have made changes accordingly in the revised manuscript. My point-by-point responses are explained in detail below. In addition, changes to the revised manuscript are highlighted in blue.

Reviewer #1 (Remarks to the Author):

I appreciate very much the amendments made to the manuscript based on my review and also on those of the other reviewers. I consider the corrections and additions sufficient and suggest a publication of the paper as it is.

Response: Thanks to the Reviewer for suggesting to publish my paper.

Reviewer #2 (Remarks to the Author):

Two additional comments are offered:

(1) The use of "I" is rather awkward. This can be easily rectified with a passive voice.

Response: The use of "I" has been avoided in the revised manuscript, most of which are rectified with a passive voice as the Reviewer suggested.

(2) The author only estimated the deformation/strain rate of Au nanowires based on lattice diffusion equation. What about Ag, Cu, and Pt? It is important to have a self-consistent story.

Response: I appreciate the Reviewer for such an important comment, which helps to enhance the article rigor and persuasiveness. I calculated strain rates for Ag and Cu in the experiments as following (See also pages 10-11): "Fig. 5c shows replicated Cu nanorod arrays at 550°C, with aspect ratios lower than those of Ag nanowires by 2-3 orders of magnitude, which is understandable if estimating their strain rate difference. By substituting $\sigma_s \sim \bar{\sigma}_s \sim 500$ MPa in the experiments, $D_s = 5090$ nm² s⁻¹, $\mu = 22$ GPa and $b = 0.286$ nm for Ag at 700°C⁴⁴, $D_s = 6$ nm² s⁻¹, $\mu = 38$ GPa and $b = 0.256$ nm for Cu at 550°C⁴⁴ into eq. 4, we obtain $\dot{\epsilon} = 12 \times 10^{-3}$ s⁻¹ for nanoimprinting of Cu, which is similar to that of replicating Au nanorods but lower than the calculated strain rate for Ag by ~ 3 order of magnitude, where $\dot{\epsilon} = 28$ s⁻¹. The consistence between experiments and theoretical calculations indicate again that lattice diffusion is a reasonable mechanism for nanoimprinting of crystalline metals well below T_m ."

However, the calculation of strain rate for nanoimprinting of Pt is not obtainable since the lacking of necessary experimental data in the literatures.